# Influence of different data-averaging methods on mean values of selected variables derived from preoperative cardiopulmonary exercise testing in patients scheduled for colorectal surgery

Ruud F. W. Franssen[1,2]*, Bart H. E. Sanders[3], Tim Takken[4], F. Jeroen Vogelaar[2], Maryska L. G. Janssen-Heijnen[2,5], Bart C. Bongers[6,7]

1 Department of Clinical Physical Therapy, VieCuri Medical Center, Venlo, The Netherlands, 2 Department of Epidemiology, GROW School for Oncology and Reproduction, Faculty of Health, Medicine and Life Sciences, Maastricht University, Maastricht, The Netherlands, 3 Department of Sports Medicine, VieCuri Medical Center, Venlo, The Netherlands, 4 Child Development and Exercise Center, Wilhelmina Children's Hospital, University Medical Center Utrecht, Utrecht, The Netherlands, 5 Department of Epidemiology, VieCuri Medical Center, Venlo, The Netherlands, 6 Department of Nutrition and Movement Sciences, Nutrition and Translational Research in Metabolism (NUTRIM), Faculty of Health, Medicine and Life Sciences, Maastricht University, Maastricht, The Netherlands, 7 Department of Surgery, Nutrition and Translational Research in Metabolism (NUTRIM), Faculty of Health, Medicine and Life Sciences, Maastricht University, Maastricht, The Netherlands

* rfranssen@viecuri.nl

## Abstract

### Introduction

Patients with a low cardiorespiratory fitness (CRF) undergoing colorectal cancer surgery have a high risk for postoperative complications. Cardiopulmonary exercise testing (CPET) to assess CRF is the gold standard for preoperative risk assessment. To aid interpretation of raw breath-by-breath data, different methods of data-averaging can be applied. This study aimed to investigate the influence of different data-averaging intervals on CPET variables used for preoperative risk assessment, as well as to evaluate whether different data-averaging intervals influence preoperative risk assessment.

### Methods

A total of 21 preoperative CPETs were interpreted by two exercise physiologists using stationary time-based data-averaging intervals of 10, 20, and 30 seconds and rolling average intervals of 3 and 7 breaths. Mean values of CPET variables between different data averaging intervals were compared using repeated measures ANOVA. The variables of interest were oxygen uptake at peak exercise ($VO_{2peak}$), oxygen uptake at the ventilatory anaerobic threshold ($VO_{2VAT}$), oxygen uptake efficiency slope (OUES), the ventilatory equivalent for carbon dioxide at the ventilatory anaerobic threshold ($VE/VCO_{2VAT}$), and the slope of the relationship between the minute ventilation and carbon dioxide production ($VE/VCO_2$-slope).

**Data Availability Statement:** All relevant data are within the paper and its Supporting Information files.

**Funding:** This study was unconditionally financed by the Research and Innovation fund VieCuri Medical Center under reference number E.21.32.004-2. The funders had no role in study design, data collection and analysis, decision to publish, or preparation of the manuscript.

**Competing interests:** The authors have declared that no competing interests exist.

## Results

Between data-averaging intervals, no statistically significant differences were found in the mean values of CPET variables except for the ventilatory equivalent for carbon dioxide at the ventilatory anaerobic threshold ($P = 0.001$). No statistically significant differences were found in the proportion of patients classified as high or low risk regardless of the used data-averaging interval.

## Conclusion

There appears to be no significant or clinically relevant influence of the evaluated data-averaging intervals on the mean values of CPET outcomes used for preoperative risk assessment. Clinicians may choose a data-averaging interval that is appropriate for optimal interpretation and data visualization of the preoperative CPET. Nevertheless, caution should be taken as the chosen data-averaging interval might lead to substantial within-patient variation for individual patients.

## Clinical trial registration

Prospectively registered at ClinicalTrials.gov (NCT05353127).

## Introduction

Preoperative aerobic fitness is independently associated with postoperative outcomes following major abdominal surgery [1]. Consequently, cardiopulmonary exercise testing (CPET) is increasingly used within multimodal preoperative risk assessment [2], as it provides an objective, non-invasive, and accurate evaluation of a patient's aerobic fitness that represents the capacity to meet the increased oxygen demand following major abdominal surgery [3, 4]. The advantage of CPET over other risk assessment tools is that CPET encompasses an integrative evaluation of the cardiovascular, pulmonary, and muscular system [5]. In addition, CPET can be used to inform collaborative decision-making, to optimize comorbidities, to triage perioperative care (e.g., ward, intensive care), to advice on preoperative physical exercise training (e.g., risk assessment, contraindications), and to guide and personalize subsequent physical exercise training prescription [6].

During CPET, a patient exercises against a progressively increasing work rate until volitional exhaustion, while breath-by-breath respiratory gasses are analyzed. The large number of data-points that are collected by the breath-by-breath sampling rate can be a challenge for data visualization, as the signal can have high variability. Therefore, data-averaging is performed to optimize graphical data display and to aid CPET interpretation (see Fig 1). Although it is generally accepted that data-averaging methods influence the numerical value of CPET-derived variables, there is no consensus among existing guidelines on the best averaging method [7].

The most frequently used CPET-derived variables that are associated with postoperative complications in the current literature are the oxygen uptake at peak exercise ($VO_{2peak}$), the oxygen uptake at the ventilatory anaerobic threshold ($VO_{2VAT}$) [2, 6, 8], and the ventilatory equivalent for carbon dioxide at the ventilatory anaerobic threshold ($VE/VCO_{2VAT}$) [9]. Measures that are less frequently used are the slope of the relationship between the minute ventilation and carbon dioxide production ($VE/VCO_2$-slope), that can be used as an alternative for the $VE/VCO_{2VAT}$ if the VAT is undeterminable [8], and the oxygen uptake efficiency slope (OUES) [10].

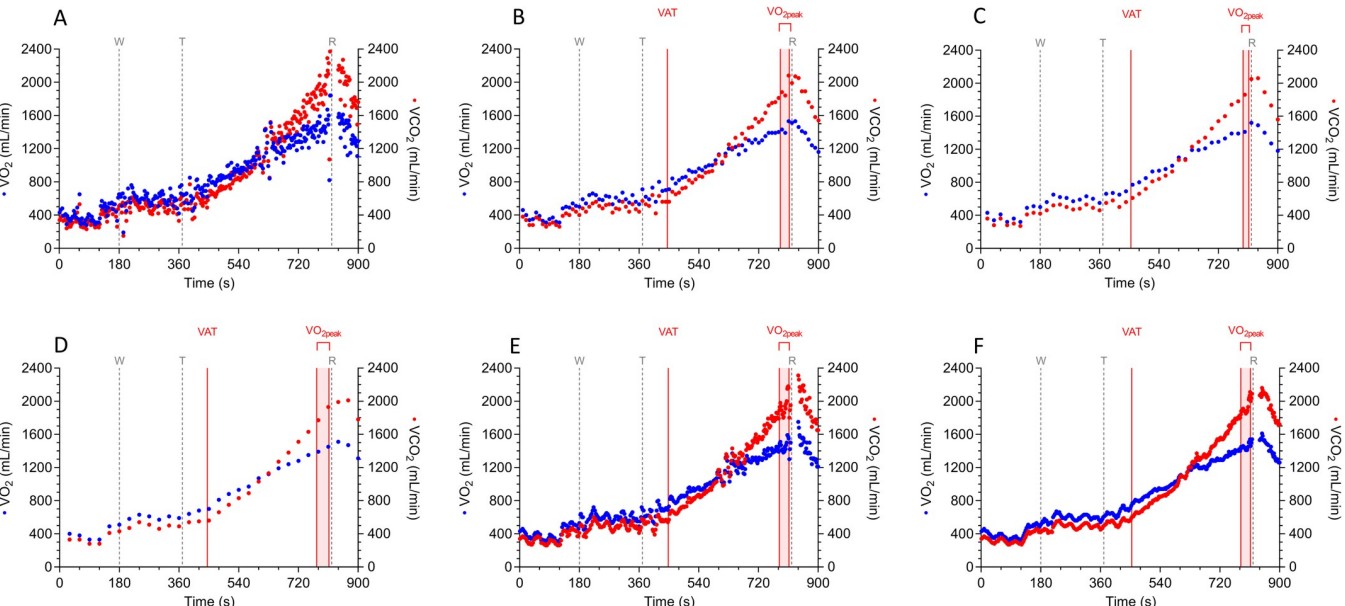

**Fig 1.** Visualization of the plot with oxygen uptake ($VO_2$) and carbon dioxide production ($VCO_2$) over time without data-averaging (graph A) and using the five different data-averaging intervals: a stationary time-based average of 10 seconds (graph B), 20 seconds (graph C), and 30 seconds (graph D), a rolling average interval of 3 breaths (graph E) and 7 breaths (graph F) in patient 21. See S1 File for a graphical display of the Wasserman plots of patient 21 with the different data-averaging intervals. Note that the number of data points is lower when stationary time-based averaging is used (and decreasing with longer data-averaging intervals) compared to when a rolling average is used. In addition, a lower number of data points leads to smoothing of the $VO_2$ and $VCO_2$ curves. Abbreviations: VAT = ventilatory anaerobic threshold; $VCO_2$ = carbon dioxide production; $VO_2$ = oxygen uptake; $VO_{2peak}$ = oxygen uptake at peak exercise. Vertical grey dotted lines represent start of the warm-up phase (W), test phase (T), and recovery phase (R).

Although preoperative risk assessment should be multimodal, CPET-derived thresholds are often used to recognize patients with a low aerobic fitness who have a high risk for adverse surgical outcomes. In major abdominal surgery, often used thresholds to identify patients at high-risk for postoperative complications are a VO2peak <18.2 mL/kg/min and/or a VO2VAT <11.1 mL/kg/min [9]. Studies in healthy individuals have shown that the numerical value of the $VO_{2peak}$ can differ as much as ~10% depending on the data-averaging method [11–13], indicating that data-averaging might significantly influence threshold determination and subsequently might affect preoperative risk assessment. To date however, there are no studies quantifying the extent to which differences in data-averaging influence the numerical value of preoperative CPET-derived variables such as $VO_{2peak}$, $VO_{2VAT}$, OUES, $VE/VCO_{2VAT}$, and $VE/VCO_2$-slope. Therefore, the primary aim of this study was to investigate the influence of different CPET data-averaging intervals on the numerical values of CPET-derived variables used for preoperative risk assessment in patients scheduled for elective colorectal cancer surgery. The secondary aim was to elucidate the impact of data-averaging intervals on the classification of patients into low or high risk for postoperative complications based on known risk assessment thresholds.

## Methods

This observational cross-sectional study was performed at the VieCuri Medical Center, a large teaching hospital in Venlo, the Netherlands. The current study was executed as a secondary analysis of data collected in a study [14] that was approved by the Medical Ethics Review Committee–Zuyderland/Zuyd (Heerlen, the Netherlands) under reference number METCZ20190150. Reporting was done using the STROBE guidelines for reporting of cross-

sectional studies [15]. The study protocol was prospectively registered at ClinicalTrials.gov (NCT05353127).

## Participants

Data from consecutive patients considered for colorectal cancer surgery who were ≥18 years of age, had a score ≤7 metabolic equivalents of task on the veterans-specific activity question-naire, and therefore performed preoperative CPET as a part of a tele-prehabilitation study [14], were collected between July 2020 and September 2021. All patients signed informed consent. Preoperative CPET was conducted after diagnosis and before any intervention or treatment was initiated.

## Preoperative cardiopulmonary exercise testing

Patients preoperatively performed incremental CPET up to volitional exertion in upright position on an electronically-braked cycle ergometer (Lode Corival, Lode BV, Groningen, the Netherlands). Prior to the test, patients were asked to refrain from vigorous physical activity, caffeine, and tobacco for 24 hours and meals for 2 hours, but to continue medication as usual. Seat height was adjusted to the participant's leg length. Before commencing CPET, forced vital capacity and forced expiratory volume in one second was obtained from maximal flow-volume curves (Ergostik, Geratherm Respiratory, Bad Kissingen, Germany) according to ATS/ERS standards [5]. Subsequently, baseline cardiopulmonary values were assessed during a three-minute rest period while seated at the cycle ergometer, thereafter a three-minute warm-up phase took place that consisted of unloaded cycling. After the warm-up, work rate was increased by constant increments of 5, 10, 15, 20, or 25 W/min in a ramp-like manner, depending on the subject's estimated physical fitness level and aimed at reaching a maximal effort within eight to twelve minutes. Throughout CPET, subjects maintained a pedaling frequency between 60 and 80 revolutions/min. The protocol continued until the patient's pedaling frequency fell definitely below 60 revolutions/min, despite strong verbal encouragement, or when the patient met the criteria for exercise termination before symptom limitation as proposed in the ATS/ACCP statement on cardiopulmonary exercise testing [5].

During CPET, subjects breathed through a facemask (Hans Rudolph, Kansas City, MO, USA) connected to an ergospirometry system (Ergostik, Geratherm Respiratory, Bad Kissingen, Germany). Before every test, calibration for respiratory gas analysis measurements (ambient air and a gas mixture of 16% oxygen and 5% carbon dioxide) and volume measurements (three-liter syringe) took place. Expired gas passed through a flow meter (triple V volume transducer), an oxygen analyzer, and a carbon dioxide analyzer. The flow meter and gas analyzers were connected to a computer that calculated breath-by-breath minute ventilation, oxygen uptake, carbon dioxide production, and the respiratory exchange ratio. Raw unfiltered breath-by-breath data was retrogradely averaged over five different data display intervals.

## Procedures

Preoperative CPET patient data was anonymized and patient characteristics other than anthropometric measures were concealed. A medical and clinical exercise physiologist (BB) and a clinical exercise physiologist (RF) determined $VO_{2peak}$, $VO_{2VAT}$, OUES, $VE/VCO_{2VAT}$, and the $VE/VCO_2$-slope in all CPETs by means of a predefined set of guidelines (see S2 File). A $VO_{2peak}$ was conceived "valid" when objective criteria for maximal volitional exertion were reached defined as an RER ≥1.10 or reaching ≥95% of the predicted maximal heart rate at peak exercise. CPET interpretation was performed using Blue Cherry software version 1.3.3.3 (Geratherm Respiratory GmbH, Bad Kissingen, Germany), in which observers interpreted the

CPET data together using TeamViewer software (TeamViewer GmbH, Göppingen, Germany). Final determination was based on consensus between the two observers. If the two observers were unable to reach consensus, a third observer (TT) was consulted. Data-averaging-intervals used were stationary time-based averages, calculated by averaging the breath-by-breath data over 10, 20, or 30 seconds and rolling averages calculated by averaging a fixed number of single breath measurements (i.e., 3 and 7), then discarding the first breath and adding a new breath to obtain a new breath averaging block. Determination of the aforementioned CPET variables was repeated for all five different data-averaging intervals.

Apart from the CPET data, the preoperative patient characteristics age, sex, body mass index, smoking status (never, former, current), age-adjusted Charlson comorbidity index, American Society of Anesthesiologists classification, veterans-specific activity questionnaire score, hemoglobin levels (mmol/L), and tumor location were recorded to characterize the study population.

### Sample size

A sample size calculation was performed with G*Power [16] for F-test repeated measures within factors. Based on a mean ± standard deviation (SD) value for $VO_{2VAT}$ of $9.7 \pm 2.3$ mL/kg/min (based on preliminary analysis of the used data) for a mean difference between data-averaging methods of minimally 0.7 mL/kg/min, the estimated effect size is estimated at ~0.30. With an α of 0.05 and a β of 0.80, a minimum of 15 CPETs are needed to detect the estimated effect size.

### Statistical analysis

Continuous data were checked for normality using the Shapiro-Wilks test. To assess the difference between different CPET data-averaging intervals, differences in mean numerical values of $VO_{2peak}$, $VO_{2VAT}$, OUES, $VE/VCO_{2VAT}$, and the $VE/VCO_2$-slope, between different data-averaging intervals were calculated and analyzed by means of within-factors repeated-measures analysis of variance (ANOVA). In case of a statistically significant difference between methods (P<0.05), post-hoc testing was performed using the Bonferroni correction to identify exact differences. Effect sizes were estimated by calculating the eta squared (i.e., sum of squares of the effect divided by the total sum of squares). To evaluate the influence of data-averaging intervals on preoperative risk assessment, individual numerical values for $VO_{2peak}$, $VO_{2VAT}$, OUES, and $VE/VCO_{2VAT}$ were compared with known preoperative risk assessment thresholds. Patients were classified as high-risk when having a $VO_{2peak}$ <18.2 mL/kg/min [9], $VO_{2VAT}$ <11.1 mL/kg/min [9], OUES/kg <20.6 [10], and/or $VE/VCO_{2VAT}$ >30.9 [9]. Cochrane's Q-test was used to determine whether differences in preoperative risk assessment exist between data-averaging methods. Differences between data-averaging methods were assumed statistically significant when P<0.05.

### Results

A total of 21 CPETs of patients with colorectal cancer (see Table 1 for patient characteristics) were re-assessed using five different data-averaging intervals. Thus, a total of 105 CPETs (five data-averaging intervals × 21 CPETs) were evaluated. Mean ± SD duration of the CPET ramp phase was $586 \pm 174$ seconds (9:46 ± 2:54 min). A valid $VO_{2peak}$ was reached in 70 (67.7%) of the evaluated CPETs. $VO_{2VAT}$ and $VE/VCO_{2VAT}$ were determinable in 104 out of 105 CPETs (99%). The OUES and $VE$-$VCO_2$-slope were determinable in all 105 CPETs.

Mean values of the CPET-derived variables ranged from 14.5 mL/kg/min to 14.6 mL/kg/min for $VO_{2peak}$, from 9.3 mL/kg/min to 9.7 mL/kg/min for $VO_{2VAT}$, from 19.1 to 19.4 for

**Table 1. Baseline characteristics of subjects.**

| Characteristics | n = 21 |
|---|---|
| Age (years) | 70.5 ± 12.5 |
| Sex ratio (male; female) | 12 (57%); 9 (43%) |
| Body mass index (kg/m$^2$) | 28.6 ± 4.9 |
| Age-adjusted Charlson comorbidity index | |
| ≤3 | 10 (47.6%) |
| 4–5 | 10 (47.6%) |
| 6+ | 1 (4.8%) |
| ASA-classification | |
| I | 4 (19.0%) |
| II | 7 (33.3%) |
| III | 9 (42.9%) |
| IV | 1 (4.8%) |
| Hemoglobin level (mmol/L) | 7.4 ± 1.2 |
| Tumor location | |
| Colon | 15 (71.4%) |
| Rectum | 6 (28.6%) |

Data are presented as mean ± standard deviation (SD) or as number (%).

OUES/kg, from 31.2 to 31.9 for VE/VCO$_{2VAT}$, and from 33.6 to 35.3 for VE/VCO$_2$-slope, dependent on the different data-averaging intervals. There was a significant difference in mean values of VO$_{2peak}$ between groups with different data averaging intervals, but this difference did not remain significant after post-hoc testing. For the variable VE/VCO$_{2VAT}$, the 3 breaths rolling average interval was statistically significant different from the time-based 20 seconds (P = 0.004) and 30 seconds (P = 0.005) data-averaging interval, as well as from the rolling average of 7 breaths (P = 0.021; see Table 2). The effect sizes for all variables were ≤0.009.

Fig 2 depicts within-patient variation in the numerical value of several CPET-derived variables using the five different data-averaging intervals. Although the numerical values for VO$_{2peak}$ were consistent (maximal within patient difference, 0.4 mL/kg/min, or 5.6%), within patient variation could be as much as 4.0 mL/kg/min for VO$_{2VAT}$ (40.8%), 5.7 for the OUES/kg (40.3%), 4.7 for VE/VCO$_{2VAT}$ (13.4%), and 10.4 (37.3%) for VE/VCO$_2$-slope when using different data-averaging intervals (see Fig 2).

When dichotomizing patients into the high or low risk category for postoperative complications based on the numerical values of each CPET variable, the proportion of patients with a high risk based on their VO$_{2peak}$ ranged from 76% to 81%, depending on the used data-averaging interval. Based on VO$_{2VAT}$ the proportion of high-risk patients ranged from 67% to 76%, whereas this ranged from 57% to 67% for OUES/kg and from 76% to 86% for VE/VCO$_{2VAT}$. As depicted in Table 3, there were no statistically significant differences in the proportion of patients who were classified as at high risk between different data-averaging-intervals. As depicted in Fig 2, individual values of some patients crossed the risk threshold depending on the data-averaging interval that was used. Based on within-patient variation, the estimated risk could differ for 1 patient when based on VO$_{2peak}$ (patient 15), for 5 patients based on VO$_{2VAT}$ (patients 1, 6, 7, 12, and 19), for 2 patients based on OUES (patients 2 and 18), and for 4 patients based on VE/VCO$_{2VAT}$ (patients 3, 7, 13, and 16), depending on the used data-averaging interval.

**Table 2. Numerical values of CPET variables using different data-averaging intervals.**

| | Data-averaging interval | | | | | P-value[a] |
|---|---|---|---|---|---|---|
| | Stationary time-based average | | | Rolling average | | |
| | 10 seconds | 20 seconds | 30 seconds | 3 breaths | 7 breaths | |
| $VO_{2peak}$ (mL/min) | 1202 (1008–1396) | 1194 (999–1389) | 1193 (997–1390) | 1201 (1008–1394) | 1200 (1005–1396) | **0.040**[c] |
| $VO_{2peak}$ (mL/kg/min) | 14.6 (12.5–16.7) | 14.5 (12.4–16.6) | 14.5 (12.4–16.6) | 14.6 (12.5–16.7) | 14.6 (12.5–16.7) | **0.012**[c] |
| Valid $VO_{2peak}$ (mL/kg/min)[b] | 16.2 (13.7–18.7) | 16.2 (13.6–18.7) | 16.1 (13.5–18.8) | 16.2 (13.6–18.8) | 16.3 (13.7–18.7) | 0.104 |
| $VO_{2VAT}$ (mL/min) | 800 (684–916) | 775 (656–895) | 764 (669–859) | 776 (668–884) | 761 (669–851) | 0.345 |
| $VO_{2VAT}$ (mL/kg/min) | 9.7 (8.5–10.9) | 9.4 (8.1–10.7) | 9.3 (8.2–10.4) | 9.5 (8.3–10.6) | 9.3 (8.2–10.4) | 0.435 |
| OUES | 1559 (1322–1795) | 1559 (1338–1779) | 1565 (1338–1792) | 1582 (1354–1809) | 1574 (1351–1798) | 0.463 |
| OUES/kg | 19.1 (16.6–21.7) | 19.2 (16.7–21.7) | 19.3 (16.7–21.8) | 19.5 (17.0–21.9) | 19.4 (16.9–21.8) | 0.479 |
| $VE/VCO_{2VAT}$ | 34.2 (31.9–36.5) | 34.6 (32.3–37.0)[d] | 35.1 (32.6–37.5)[d] | 33.6 (31.4–35.8)[d] | 34.4 (32.0–36.8)[d] | **0.001** |
| $VE/VCO_2$-slope | 31.4 (28.6–34.1) | 31.8 (28.8–35.0) | 31.2 (28.4–34.0) | 31.8 (28.9–34.7) | 31.8 (29.0–34.7) | 0.608 |

Data are presented as mean and 95% confidence interval (CI), unless stated otherwise.

Abbreviations: OUES = oxygen uptake efficiency slope; $VE/VCO_2$-slope = the slope of the relationship between the minute ventilation and carbon dioxide production; $VE/VCO_{2VAT}$ = ventilatory equivalent for carbon dioxide at the ventilatory anaerobic threshold; $VO_{2peak}$ = oxygen uptake at peak exercise; $VO_{2VAT}$ = oxygen uptake at the ventilatory anaerobic threshold.

[a]: as determined by repeated-measures ANOVA (within factors).

[b]: as determined by a respiratory exchange ratio at peak exercise $\geq 1.10$ and/or a heart rate at peak exercise $\geq 95\%$ of the predicted maximal heart rate based on the formula $208 - (0.8 \times age$ in years).

[c]: did not remain significant after post hoc testing with Bonferroni correction.

[d]: the 3 breaths rolling average interval was statistically significant different from the stationary time-based interval of 20 seconds (P = 0.004) and 30 seconds (P = 0.005), as well as from the 7 breaths rolling average interval (P = 0.021).

## Discussion

To our knowledge, the current study was the first study that aimed to investigate whether the selection of different CPET data-averaging intervals would translate into differences in mean values of CPET-derived variables in patients with colorectal cancer who performed CPET for preoperative risk assessment. As CPET-derived variables are used to preoperatively classify patients into having a low or high risk for postoperative complications based on their CRF, the secondary aim of the current study was to investigate whether potential differences in the numerical values of CPET-derived variables would lead to differences in preoperative risk classification. Based on the mean values of the CPET-derived variables there were only statistically significant differences for the variables $VO_{2peak}$ and $VE/VCO_{2VAT}$ between different data-averaging intervals. For $VO_{2peak}$, the between-group difference did not remain significant after post-hoc analysis, whereas data-averaging group differences $VE/VCO_{2VAT}$ were statistically significant between the 3 breaths moving average and the 20- and 30-second time-based interval, as well as the 7 breaths moving average.

For $VO_{2peak}$, the greatest observed difference between data-averaging groups was 0.1 mL/kg/min. Given that the coefficient of variation (a measure of reproducibility) for $VO_{2peak}$ is estimated to be between ~5% and ~9% [5] (i.e., between ~0.7 mL/kg/min and ~1.3 mL/kg/min based on mean values of $VO_{2peak}$ in the current study), the observed maximal difference of 0.1 mL/kg/min is not clinically relevant. The observation that this small difference in $VO_{2peak}$ is not clinically relevant is further emphasized by the fact that no differences were found between the proportion of patients who were classified as low or high risk based on $VO_{2peak}$ when using different data-averaging intervals in the current study. Provided that the critical difference of $VE/VCO_{2VAT}$ in patients with colorectal cancer is assumed to be ~10% [17], the

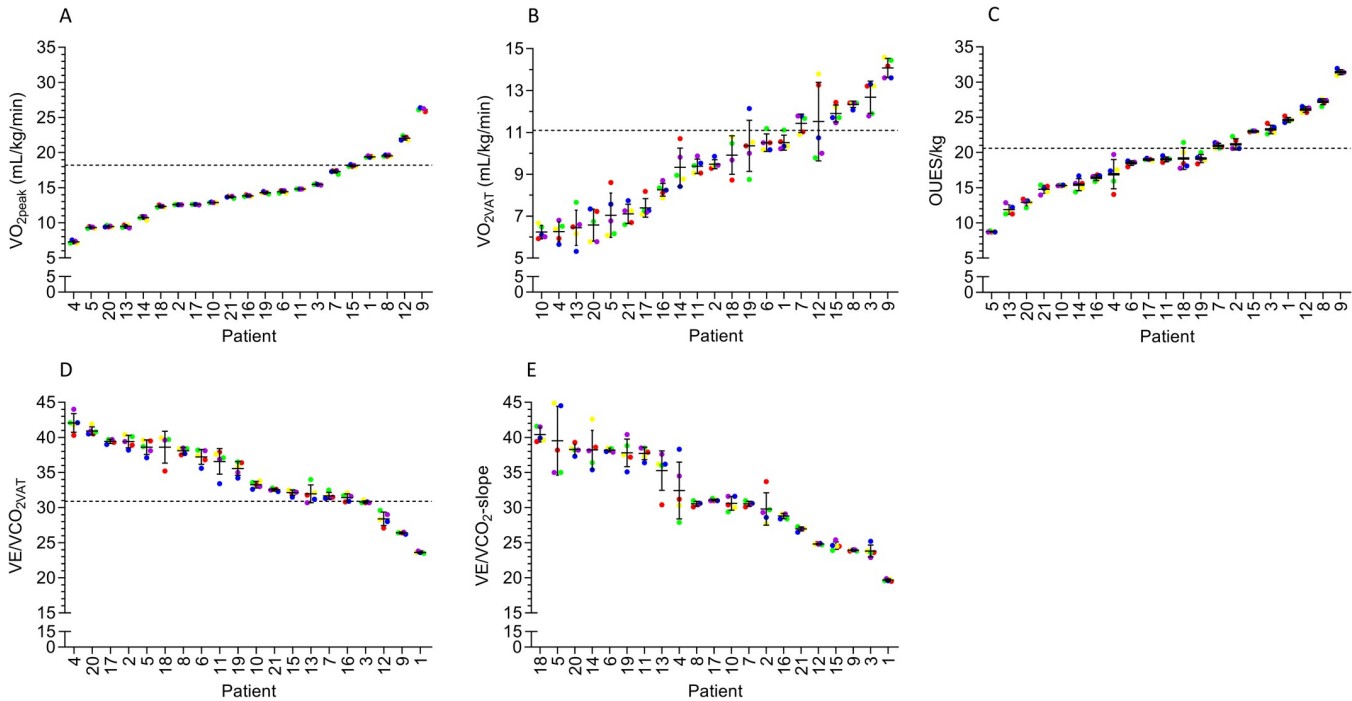

**Fig 2.** Variation in the observed values of $VO_{2peak}$ (graph A), $VO_{2VAT}$ (graph B), OUES (graph C), $VE/VCO_{2VAT}$ (graph D), and the $VE/VCO_2$-slope (graph E) within individual patients. Dots represent individual numerical value with a unique color for each data-averaging interval throughout the graphs (red = 10 seconds; yellow = 20 seconds; green = 30 seconds; blue = 3 breaths; purple = 7 breaths). Error bars represent the mean values and 95% confidence intervals. Horizontal dotted lines represent known risk assessment thresholds defined as 18.2 mL/kg/min for $VO_{2peak}$ (graph A), 11.1 mL/kg/min for $VO_{2VAT}$ (graph B), <20.6 for OUES (graph C), and >30.9 for $VE/VCO_{2VAT}$ (graph D). Note that individual values of patients often cross the risk threshold (dotted horizontal line). These patients might have a different risk estimation depending on the data-averaging interval. Abbreviations: OUES = oxygen uptake efficiency slope; $VE/VCO_2$-slope = the slope of the relationship between the minute ventilation and carbon dioxide production; $VE/VCO_{2VAT}$ = ventilatory equivalent for carbon dioxide at the ventilatory anaerobic threshold; $VO_{2peak}$ = oxygen uptake at peak exercise; $VO_{2VAT}$ = oxygen uptake at the ventilatory anaerobic threshold.

maximal mean difference of 1.5 (5%) measured in the current study is not deemed clinically relevant. The observation that differences in the mean values of the $VE/VCO_2$-slope are not clinically relevant is also supported by the very small effect size (0.009).

The main purpose of using data-averaging of CPET data is to reduce noise of breath-by-breath fluctuations and to aid CPET interpretation [5]. In the current study there seem to be

**Table 3. Effect of different data-averaging intervals on classifying patients as having a high-risk for postoperative complications.**

| | Data-averaging interval | | | | | |
| --- | --- | --- | --- | --- | --- | --- |
| | Stationary time-based average | | | Rolling average | | |
| | 10 seconds, n (%) | 20 seconds n (%) | 30 seconds (%) n (%) | 3 breaths n (%) | 7 breaths n (%) | P-value[a] |
| $VO_{2peak}$ | 17 (81%) | 17 (81%) | 17 (81%) | 17 (81%) | 16 (76%) | 0.406 |
| $VO_{2VAT}$ | 16 (76%) | 16 (76%) | 14 (67%) | 14 (67%) | 16 (76%) | 0.615 |
| OUES/kg | 13 (62%) | 13 (62%) | 12 (57%) | 14 (67%) | 14 (67%) | 0.231 |
| $VE/VCO_{2VAT}$ | 16 (76%) | 18 (86%) | 17 (81%) | 15 (71%) | 16 (76%) | 0.334 |

Data are presented as number (%).

Abbreviations: OUES = oxygen uptake efficiency slope; $VE/VCO_{2VAT}$ = ventilatory equivalent for carbon dioxide at the ventilatory anaerobic threshold; $VO_{2peak}$ = oxygen uptake at peak exercise; $VO_{2VAT}$ = oxygen uptake at the ventilatory anaerobic threshold.

[a]: determined by Cochrane's Q-test.

no clinically relevant differences in CPET-derived variables between different data-averaging intervals. This is a reassuring observation that opens possibilities to be flexible in the use of data-averaging intervals as long as the interval is within certain boundaries. That is, the type and duration of CPET can be taken into consideration when determining the optimal data-averaging interval [7]. For example, using longer averaging intervals in longer tests, or using a rolling average for noisy data. On the other hand, longer intervals might mask dynamic patho-physiological processes such oscillatory breathing. In these circumstances shorter time-based intervals might be optimal [7]. For preoperative exercise testing a stationary time-based average of 10 seconds, or a breath-based rolling average of 3 or 7 seconds might provide a good trade-off between de number of data-points and the duration of the test.

Although the literature is scarce with regard to the influence of data-averaging intervals on the determination of CPET-derived variables (and only available for $VO_{2peak}$), results of the current study are in line with a previous publication in which the effect of data-averaging intervals on $VO_{2peak}$ in 22 healthy athletic subjects was investigated [18]. The authors found that only a stationary time-based data-averaging interval of 60 seconds was significantly different from all other data-averaging intervals (10, 15, 20, and 30 seconds) [18]. In a study evaluating $VO_{2peak}$ values of 15 patients who were screened for heart transplant surgery (with comparable mean $VO_{2peak}$ values as observed in the current study), no significant differences were found between stationary time-based data-averaging intervals of 15 and 30 seconds, and a 8 breaths rolling average interval [19]. Moreover, only a 60-second stationary time-based data-averaging interval was statistically significantly different from the aforementioned data-averaging intervals [19]. These long data-averaging intervals (of 60 seconds or more) are probably not used very often in preoperative CPETs and are not recommend by current preoperative CPET guidelines [8].

Based on the results of this study, the recommendation in the preoperative CPET guideline to use a breath-based data-averaging interval of 3–5 breaths or a time-based data-averaging interval of ~20 seconds seems plausible when evaluating the mean (group level) values. Nevertheless, caution should be taken when evaluating individual patients, as different data-averaging intervals caused substantial variation in the numerical values of CPET-derived variables within patients. As depicted in Fig 2, individual values of patients could differ as much as ~40%. In individual patients, the chosen data-averaging interval could induce a shift of that patient from low to high risk or vice versa. This is an important observation, as risk assessment could influence surgical planning for individual patients (e.g., enrollment in prehabilitation program, referring to a higher care unit postoperatively) and the shared decision-making process. It is recognized that preoperative risk assessment is not solely based on risk thresholds determined by CPET, but rather consists of a composite assessment, taking into account the full CPET in combination with other preoperative risk factors such as, but not limited to, malnutrition, comorbidities, and geriatric status. Nevertheless, the influence of the data-averaging interval could be taken into consideration, especially in patients in which the CPET values are close to the risk classification cut-off point. In addition, instead of rigid cut-off points inducing black and white risk assessment, grey zones (intermediate risk) could be introduced to account for individual differences [17].

A limitation of the current study was that $VO_{2peak}$ was determined over a ~30 second interval [5] regardless of the data-averaging interval that was used. The use of the fixed 30 second interval might have masked some of the variability caused by the data-averaging interval, explaining the very small differences of $VO_{2peak}$ values between data averaging intervals. A strength of the current study is that variation other than variation coming from the data-averaging interval was minimized. Firstly, by repeating interpretation of the 21 CPETs that were retrospectively formatted using 5 different data-averaging intervals, as opposed to repeated

testing of patients with different data averaging intervals. Secondly, to account for inter-observer variability, CPET interpretation was done by two clinical exercise physiologists, based on consensus, and by using a predefined set of guidelines (see S2 File). By doing so, the observed variation between groups of data-averaging intervals was exclusively caused by the used data-averaging interval and not by within-patient biological variation, measurement error, or inter-observer variability.

The current study opens possibilities for clinicians to be flexible in the data-averaging interval that is used for interpretation of the preoperative CPET. Current CPET literature does not provide clear and consistent guidance for clinicians about the choice of a data-averaging interval [7, 8]. As different (patho)physiological patterns might require different data-visualization, future research could focus on investigating optimal data-visualization methods that best fit the aim of the CPET, the properties of the CPET, and the (patho)physiological process the clinician is willing to evaluate.

## Conclusion

On a group level there appear to be no clinically relevant differences in the mean values of $VO_{2peak}$, $VO_{2VAT}$, OUES, $VE/VCO_{2VAT}$, and $VE/VCO_2$-slope between different data-averaging intervals used for interpretation of preoperative CPET in patients with colorectal cancer. In addition, the choice of data-averaging interval does not influence the proportion of patients classified as high or low risk for complications based on their exercise tolerance. Nevertheless, the chosen data-averaging interval might lead to substantial within patient variation for individual patients and should therefore be considered in patients in which the CPET values are close to the risk classification cut-off point.

## Supporting information

**S1 File. Graphical display of the Wasserman plots of patient 21.**
(PDF)

**S2 File. Guideline for systematic interpretation of preoperative cardiopulmonary exercise testing.**
(PDF)

**S1 Dataset.**
(XLSX)

## Acknowledgments

The authors like to thank Accuramed BVBA (Halen, Belgium) for providing a free unrestricted copy of the Blue Cherry software to support this study.

## Author Contributions

**Conceptualization:** Ruud F. W. Franssen, Tim Takken, F. Jeroen Vogelaar, Maryska L. G. Janssen-Heijnen, Bart C. Bongers.

**Data curation:** Ruud F. W. Franssen, Bart H. E. Sanders.

**Formal analysis:** Ruud F. W. Franssen, Bart H. E. Sanders, Maryska L. G. Janssen-Heijnen, Bart C. Bongers.

**Funding acquisition:** Ruud F. W. Franssen, F. Jeroen Vogelaar, Maryska L. G. Janssen-Heijnen, Bart C. Bongers.

**Investigation:** Ruud F. W. Franssen, Bart C. Bongers.

**Methodology:** Ruud F. W. Franssen, Bart H. E. Sanders, Tim Takken, F. Jeroen Vogelaar, Maryska L. G. Janssen-Heijnen, Bart C. Bongers.

**Project administration:** Ruud F. W. Franssen, Bart H. E. Sanders.

**Resources:** Ruud F. W. Franssen.

**Software:** Ruud F. W. Franssen.

**Supervision:** F. Jeroen Vogelaar, Maryska L. G. Janssen-Heijnen, Bart C. Bongers.

**Visualization:** Ruud F. W. Franssen, Tim Takken, Bart C. Bongers.

**Writing – original draft:** Ruud F. W. Franssen, Bart H. E. Sanders, Bart C. Bongers.

**Writing – review & editing:** Ruud F. W. Franssen, Bart H. E. Sanders, Tim Takken, F. Jeroen Vogelaar, Maryska L. G. Janssen-Heijnen, Bart C. Bongers.

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
