## [Decision Letter · Decision Letter 0]

8 Nov 2022

PONE-D-22-28377Influence of different data-averaging methods on preoperative risk assessment using cardiopulmonary exercise testing in patients scheduled for colorectal surgeryPLOS ONE

Dear Dr. franssen,

Thank you for submitting your manuscript to PLOS ONE. After careful consideration, we feel that it has merit but does not fully meet PLOS ONE’s publication criteria as it currently stands. Therefore, we invite you to submit a revised version of the manuscript that addresses the points raised during the review process.

We look forward to receiving your revised manuscript.

Kind regards,

Sampath Kumar Amaravadi, Ph.D

Academic Editor

PLOS ONE

“This study was unconditionally financed by the Research and Innovation fund VieCuri Medical Center under reference number E.21.32.004-2.”

Additional Editor Comments:

Dear Author,

Kindly revise the manuscript with reviewer comments and submit the same

Reviewers' comments:

Reviewer's Responses to Questions

**Comments to the Author**

1. Is the manuscript technically sound, and do the data support the conclusions?

Reviewer #1: Yes

Reviewer #2: No

2. Has the statistical analysis been performed appropriately and rigorously? 

Reviewer #1: Yes

Reviewer #2: No

3. Have the authors made all data underlying the findings in their manuscript fully available?

Reviewer #1: Yes

Reviewer #2: No

4. Is the manuscript presented in an intelligible fashion and written in standard English?

Reviewer #1: Yes

Reviewer #2: Yes

5. Review Comments to the Author

Reviewer #1: Interesting substudy of an approved study. appropriately conducted analysis

A few points to address :

The authors write as if assuming a degree of precision to interpretation, and hard differentation into low or high risk categories which isn`t typical of real life practice .

The authors have referenced the Rose “Grey Zone” paper

Rose GA, Davies RG, Davison GW, Adams RA, Williams IM, Lewis MH, et al. The 434 cardiopulmonary exercise test grey zone; optimising fitness stratification by application of critical 435 difference. Br J Anaesth. 2018;120(6):1187-94

…which makes this point well . The authors of the current study similarly make it clear in their conclusion that any apparent differences in values due to averaging are not clinically relevant . Important that they convey this clearly in the abstract

Patients are generally not classified in a binary fashion as either high or low risk based on whether they are above or below strictly defined single threshold limits / Overall summary of risk tends to be a composite assessment taking into account many CPET measurements , & comorbidities.

The CPET is not applied as a pass or fail test . CPET derived Individualised Risk estimates are used to add weight to shared decision making about whether to proceed to surgery , and to help determine perioperative care pathway ( eg High Dependency Care versus ward based care pathway ) . It is worth covering this point about clinical application in the discussion

Section by section

Title : “ Influence of different data-averaging methods on preoperative risk assessment using

cardiopulmonary exercise testing in patients scheduled for colorectal surgery”. I think this title is over stating the impact of the study findings. More appropriate to say “ Influence of different data-averaging methods on mean values of selected variables derived from cardiopulmonary exercise testing in patients scheduled for colorectal surgery”

Abstract could define exactly which CPET derived variables were considered in the study – these are defined on p11 , para 2

Methods – clear

Results clear

Table 2 is really good to demonstrate the results

Figure 1 is a graphical representation of averaging – this is a powerful figure to illustrate the point that regardless of averaging method – the thresholds of interest are generally consistent

Figure 2 similarly is powerful to demonstrate the spread of individual patients` values dependent on the method of averaging

P 11 para 2 – several CPET derived variables defined – however the abbrevations (VO2VAT) and(VE/VCO2VAT) are unusual . I`m accustomed to VO2AT and VE/VCO2AT.

Discussion :

Re-states main findings and discusses implications in context . Reasonable and moderate conclusions

Reviewer #2: This study make nonsense. If the aim is to establish whether the averaging interval of VO2 data has an impact in the VO2max value and the medical decisions adopting depending on this value why are the authors analyzing all results using 30-s averages? Moreover, the metabolic cart data are incomplete, please add PETO2 and PETCO2 data.

Are all metabolic carts used disclosed? Or all test were performed with the same metabolic cart?

Please add as as an example the full output data of one of the incremental exercise test with breath by breath data and then the calculated 10, 20 or 30 s averaged values.

6. PLOS authors have the option to publish the peer review history of their article (what does this mean?). If published, this will include your full peer review and any attached files.

Reviewer #1: **Yes: **Gary Minto

Reviewer #2: No

---

## [Author Response · Author response to Decision Letter 0]

22 Nov 2022

Response to reviewers’ comments:

Reviewer #1: Interesting substudy of an approved study. appropriately conducted analysis

A few points to address :

The authors write as if assuming a degree of precision to interpretation, and hard differentation into low or high risk categories which isn`t typical of real life practice .

The authors have referenced the Rose “Grey Zone” paper

Rose GA, Davies RG, Davison GW, Adams RA, Williams IM, Lewis MH, et al. The 434 cardiopulmonary exercise test grey zone; optimising fitness stratification by application of critical 435 difference. Br J Anaesth. 2018;120(6):1187-94

…which makes this point well . The authors of the current study similarly make it clear in their conclusion that any apparent differences in values due to averaging are not clinically relevant. Important that they convey this clearly in the abstract

Authors’ response: We fully agree with the reviewer that preoperative risk assessment should be seen in the broader context of CPET interpretation and patient assessment in general. 

We added “or clinically relevant” to the conclusion of the abstract

To avoid any unclarities regarding the point put forward by the reviewer about the conclusion in the abstract, we also adjusted the abstract as follows: The section “the proportion of patients classified as high or low risk for complications.” was deleted. And stated more in general as “perioperative risk assessment.”

Patients are generally not classified in a binary fashion as either high or low risk based on whether they are above or below strictly defined single threshold limits / Overall summary of risk tends to be a composite assessment taking into account many CPET measurements , & comorbidities.

The CPET is not applied as a pass or fail test . CPET derived Individualised Risk estimates are used to add weight to shared decision making about whether to proceed to surgery , and to help determine perioperative care pathway ( eg High Dependency Care versus ward based care pathway ) . It is worth covering this point about clinical application in the discussion

Authors’ response: In the manuscript, this point was briefly mentioned in the discussion on Page 21, Line 334 - 335. Nevertheless, we agree that we should elaborate more on this. Therefore, we adjusted this part of the discussion on page 21 Line 335-338 as follows: 

Although it is recognized that risk assessment is not solely based on the numerical value of any CPET-derived variable

“It is recognized that preoperative risk assessment is not solely based on risk thresholds determined by CPET, but rather consists of a composite assessment, taking into account the full CPET in combination with other preoperative risk factors such as, but not limited to, malnutrition, comorbidities, and geriatric status.” 

Section by section

Title : “ Influence of different data-averaging methods on preoperative risk assessment using

cardiopulmonary exercise testing in patients scheduled for colorectal surgery”. I think this title is over stating the impact of the study findings. More appropriate to say “ Influence of different data-averaging methods on mean values of selected variables derived from cardiopulmonary exercise testing in patients scheduled for colorectal surgery”

Authors’ response: We adjusted the title according to the suggestion of the reviewer. The new title reads as follows: “Influence of different data-averaging methods on mean values of selected variables derived from preoperative cardiopulmonary exercise testing in patients scheduled for colorectal surgery” 

Abstract could define exactly which CPET derived variables were considered in the study – these are defined on p11 , para 2

Methods – clear

Results clear

Table 2 is really good to demonstrate the results

Figure 1 is a graphical representation of averaging – this is a powerful figure to illustrate the point that regardless of averaging method – the thresholds of interest are generally consistent

Figure 2 similarly is powerful to demonstrate the spread of individual patients` values dependent on the method of averaging

Authors’ response: We thank the reviewer for his positive evaluation of the manuscript and figures and the suggestion to define the exact variables in the abstract. We added the following to the methods section of the abstract: “The variables of interest were oxygen uptake at peak exercise (VO2peak), oxygen uptake at the ventilatory anaerobic threshold (VO2VAT), oxygen uptake efficiency slope (OUES), the ventilatory equivalent for carbon dioxide at the ventilatory anaerobic threshold (VE/VCO2VAT), and the slope of the relationship between the minute ventilation and carbon dioxide production (VE/VCO2-slope).” 

P 11 para 2 – several CPET derived variables defined – however the abbrevations (VO2VAT) and(VE/VCO2VAT) are unusual . I`m accustomed to VO2AT and VE/VCO2AT.

Authors’ response: Indeed, many different terminologies are used within the literature to define CPET-derived thresholds. We prefer to use the term ventilatory anaerobic threshold (VAT) over anaerobic threshold (AT), as is it makes more clear that the thresholds are derived from respiratory gasses (as opposed to i.e., lactate measurement). Nevertheless, if the reviewer or the editor insist on using AT instead, we are willing to adjust the terminology throughout the manuscript. 

Discussion :

Re-states main findings and discusses implications in context . Reasonable and moderate conclusions

Reviewer #2: 

This study make nonsense. If the aim is to establish whether the averaging interval of VO2 data has an impact in the VO2max value and the medical decisions adopting depending on this value why are the authors analyzing all results using 30-s averages? 

Authors’ response: We thank the reviewer for the critical appraisal of our study but we regret that the reviewer qualifies our study as nonsense. Before we can reflect on the reviewers points, we must emphasize that we did not analyze all results using 30-second data-averaging. In addition, we did not aim to establish whether data-averaging intervals impact VO2peak values alone. Instead, we assessed many more variables (as outlined on page 10) using 5 different data averaging intervals. Therefore, the point the reviewer merely refers to the determination of VO2peak values. 

Regarding the determination of VO2peak values, let us explain why we chose to analyze VO2peak values over an interval close to 30-seconds regardless of the used data-averaging method. In most (if not all) software packages used for CPET interpretation, VO2peak is calculated over a period that has to be manually set. That is, instead of using one point that is allocated as VO2peak, a lower limit and upper limit have to be set in order to demark the period over which VO2peak is calculated (see also Figure 1, red shaded area in each graph). With this in mind, the method chosen in the current study mimics how VO2peak is estimated in clinical practice. Nevertheless, we do agree with the reviewer that by doing so, the variability introduced by the data-averaging interval is attenuated as was already discussed on page 21, line 344-347. Nevertheless, this also reflects routine practice. 

Moreover, the metabolic cart data are incomplete, please add PETO2 and PETCO2 data.

Authors’ response: We are not sure what the reviewer is referring to as we did not disclose any metabolic cart data. We did however include the raw data on which the analyses are based. Is that what the reviewer is referring to? 

Are all metabolic carts used disclosed? Or all test were performed with the same metabolic cart?

Authors’ response: All tests are performed on the same metabolic cart as reported in the methods section in the paragraph “Preoperative cardiopulmonary exercise testing” on page 7 and 8. We included a data file with the raw data that was used for the analysis. We did not include the metabolic cart data as we believe that due to the relative small study sample it would be hard to guarantee anonymity of our participants. Nevertheless, on reasonable request the metabolic cart data could be shared after personal communication with the authors. 

Please add as as an example the full output data of one of the incremental exercise test with breath by breath data and then the calculated 10, 20 or 30 s averaged values.

Authors’ response: We added an example of a graphical display of the metabolic cart of patient 21 as Supporting information S3.

---

## [Decision Letter · Decision Letter 1]

30 Jan 2023

PONE-D-22-28377R1Influence of different data-averaging methods on mean values of selected variables derived from preoperative cardiopulmonary exercise testing in patients scheduled for colorectal surgery.PLOS ONE

Dear Dr. franssen,

Thank you for submitting your manuscript to PLOS ONE. After careful consideration, we feel that it has merit but does not fully meet PLOS ONE’s publication criteria as it currently stands. Therefore, we invite you to submit a revised version of the manuscript that addresses the points raised during the review process.

 Please go through the comments of the reviewers. One reviewer has asked to change the focus and consider removing the second aim. Do consider the comment and provide a response.  Otherwise, please address the comments by reviewer 4. 

We look forward to receiving your revised manuscript.

Kind regards,

Lindsay Bottoms

Academic Editor

PLOS ONE

Journal Requirements:

Reviewers' comments:

Reviewer's Responses to Questions

**Comments to the Author**

1. If the authors have adequately addressed your comments raised in a previous round of review and you feel that this manuscript is now acceptable for publication, you may indicate that here to bypass the “Comments to the Author” section, enter your conflict of interest statement in the “Confidential to Editor” section, and submit your "Accept" recommendation.

Reviewer #1: All comments have been addressed

Reviewer #3: (No Response)

Reviewer #4: (No Response)

Reviewer #5: (No Response)

2. Is the manuscript technically sound, and do the data support the conclusions?

Reviewer #1: Yes

Reviewer #3: No

Reviewer #4: Yes

Reviewer #5: Yes

3. Has the statistical analysis been performed appropriately and rigorously? 

Reviewer #1: Yes

Reviewer #3: No

Reviewer #4: Yes

Reviewer #5: Yes

4. Have the authors made all data underlying the findings in their manuscript fully available?

Reviewer #1: Yes

Reviewer #3: Yes

Reviewer #4: Yes

Reviewer #5: (No Response)

5. Is the manuscript presented in an intelligible fashion and written in standard English?

Reviewer #1: Yes

Reviewer #3: Yes

Reviewer #4: Yes

Reviewer #5: Yes

6. Review Comments to the Author

Reviewer #1: (No Response)

Reviewer #3: The introduction is not specific to the findings of this study until line 108. Although you mention you have addressed reviewer #1 concerns about the interpretation of CPEX utility, comments like lines 106-107 are misleading. I absolutely agree that threshold values are far from agreed upon across surgical specialties. CPEX is at best a single test that should be interpreted in context. I do not believe that patients would be denied surgery on the result on this single test in many institutions - unless you have data on this. This should be made clearer in your paper, not just in the conclusion, which seems at odds now with the introduction.

I do believe that it is useful to the examine the influence of data-averaging methods for this test, which the authors have done well. This, in my opinion, should be the focus of the paper. I do not believe the secondary aim of this study is worthwhile. I understand the point being made, but clarifying the influence of different data averaging method is sufficient. We can interpret the clinical application ourselves.

I therefore recommend changing the focus of the paper to reflect the above.

Reviewer #4: General comments

The authors of this study addressed a relatively simple question, yet potentially important from a practical perspective, whether the way data from CPET is averaged affects parameters estimated from the CPET in a clinical population.

Overall, the manuscript is well written, and contains all technically relevant information.

I think the authors and previous reviewers have done a thorough job, and thus my comments remain relatively minor. I hope my comments help improve the manuscript.

The main comments (see below) refers to the interpretation of some of the results in the discussion.

Specific comments

Line 28 abstract – please abbreviate cardiorespiratory fitness as CRF.

Line 64-66 – perhaps you can make the point/link (or make this point clearer) that a key goal of prehabilitation is to increase CRF, and increases in CRF have a clinically relevant positive effect on subsequent complications.

Line 100 and throughout the study: I would like see the authors’ view on whether the term ‘anaerobic threshold’ should still be used, even if that’s still common in this field? See for a review https://pubmed.ncbi.nlm.nih.gov/33112439/. The term gas exchange threshold perhaps best captures this first threshold? You also refer in the supplementary material 2 to RCP, but this was not reported in the manuscript. Was this (RCP) only determined for the purpose of determining the VE-VCO2 slope? I fully appreciate the terminology in this field is far from standard and different labs/fields use different terms, I can see the terminology was already addressed in previous reviews, but would like to bring this up again.

Figure 1. This may be an error on my end – but the resolution of the figure is not great. Please check before publication. Same in other figures.

Line 172 – how was disagreement defined? E.g. As a 5%? 10? difference between two initials assessments?

Line 281 – Would it make sense to change from “exercise tolerance” to CRF? I would argue that CRF is then what underpins exercise tolerance – to be ability perform a task without reaching task-failure.

The discussion and conclusion should better reflect some of the results reported, in my view– specifically that “within patient variation could be as much as 4.0 mL/kg/min for VO2VAT (40.8%), 5.7 for the OUES/kg (40.3%), 4.7 for VE/VCO2VAT (13.4%), and 10.4 (37.3%) for VE/VCO2-slope when using different data-averaging intervals (see Figure 2).”

I can see this is addressed in the last part of the discussion, and also mentioned in the conclusion, but I would like to ask if this can be stressed / made more clear? For example, in the abstract you state: "nor does the choice of data-averaging interval influence

perioperative risk assessment", but the results show the opposite (at least at the individual level, which to me is more important than data at the group level in this instance), see lines 263 onwards.

Reviewer #5: My perception is that the study is very well organized in writing and the statistic methods were correctly used. However kindly allow me one point to address, as per below:

- The authors should also include the effect size in the statistics. This will give a more accurate picture of what the findings represents.

Another point to enphasize is that the discussion refers to the main finding of the study, which is very appropriate.

This type of initiative will always be very welcome since post surgical complications can be significantly affected (reduced) by the use of apropriate preoperative procedures, bearing in mind that the aim of this study is to manly investigate the methods which are presently being clinically used.

7. PLOS authors have the option to publish the peer review history of their article (what does this mean?). If published, this will include your full peer review and any attached files.

Reviewer #1: **Yes: **Gary Minto

Reviewer #3: No

Reviewer #4: **Yes: **Daniel Muniz-Pumares

Reviewer #5: No

---

## [Author Response · Author response to Decision Letter 1]

7 Feb 2023

Response to reviewers’ comments:

Reviewer #1: Interesting substudy of an approved study. appropriately conducted analysis

A few points to address :

The authors write as if assuming a degree of precision to interpretation, and hard differentation into low or high risk categories which isn`t typical of real life practice .

The authors have referenced the Rose “Grey Zone” paper

Rose GA, Davies RG, Davison GW, Adams RA, Williams IM, Lewis MH, et al. The 434 cardiopulmonary exercise test grey zone; optimising fitness stratification by application of critical 435 difference. Br J Anaesth. 2018;120(6):1187-94

…which makes this point well . The authors of the current study similarly make it clear in their conclusion that any apparent differences in values due to averaging are not clinically relevant. Important that they convey this clearly in the abstract

Authors’ response: We fully agree with the reviewer that preoperative risk assessment should be seen in the broader context of CPET interpretation and patient assessment in general. 

We added “or clinically relevant” to the conclusion of the abstract

To avoid any unclarities regarding the point put forward by the reviewer about the conclusion in the abstract, we also adjusted the abstract as follows: The section “the proportion of patients classified as high or low risk for complications.” was deleted. And stated more in general as “perioperative risk assessment.”

Patients are generally not classified in a binary fashion as either high or low risk based on whether they are above or below strictly defined single threshold limits / Overall summary of risk tends to be a composite assessment taking into account many CPET measurements , & comorbidities.

The CPET is not applied as a pass or fail test . CPET derived Individualised Risk estimates are used to add weight to shared decision making about whether to proceed to surgery , and to help determine perioperative care pathway ( eg High Dependency Care versus ward based care pathway ) . It is worth covering this point about clinical application in the discussion

Authors’ response: In the manuscript, this point was briefly mentioned in the discussion on Page 21, Line 334 - 335. Nevertheless, we agree that we should elaborate more on this. Therefore, we adjusted this part of the discussion on page 21 Line 335-338 as follows: 

Although it is recognized that risk assessment is not solely based on the numerical value of any CPET-derived variable

“It is recognized that preoperative risk assessment is not solely based on risk thresholds determined by CPET, but rather consists of a composite assessment, taking into account the full CPET in combination with other preoperative risk factors such as, but not limited to, malnutrition, comorbidities, and geriatric status.” 

Section by section

Title : “ Influence of different data-averaging methods on preoperative risk assessment using

cardiopulmonary exercise testing in patients scheduled for colorectal surgery”. I think this title is over stating the impact of the study findings. More appropriate to say “ Influence of different data-averaging methods on mean values of selected variables derived from cardiopulmonary exercise testing in patients scheduled for colorectal surgery”

Authors’ response: We adjusted the title according to the suggestion of the reviewer. The new title reads as follows: “Influence of different data-averaging methods on mean values of selected variables derived from preoperative cardiopulmonary exercise testing in patients scheduled for colorectal surgery” 

Abstract could define exactly which CPET derived variables were considered in the study – these are defined on p11 , para 2

Methods – clear

Results clear

Table 2 is really good to demonstrate the results

Figure 1 is a graphical representation of averaging – this is a powerful figure to illustrate the point that regardless of averaging method – the thresholds of interest are generally consistent

Figure 2 similarly is powerful to demonstrate the spread of individual patients` values dependent on the method of averaging

Authors’ response: We thank the reviewer for his positive evaluation of the manuscript and figures and the suggestion to define the exact variables in the abstract. We added the following to the methods section of the abstract: “The variables of interest were oxygen uptake at peak exercise (VO2peak), oxygen uptake at the ventilatory anaerobic threshold (VO2VAT), oxygen uptake efficiency slope (OUES), the ventilatory equivalent for carbon dioxide at the ventilatory anaerobic threshold (VE/VCO2VAT), and the slope of the relationship between the minute ventilation and carbon dioxide production (VE/VCO2-slope).” 

P 11 para 2 – several CPET derived variables defined – however the abbrevations (VO2VAT) and(VE/VCO2VAT) are unusual . I`m accustomed to VO2AT and VE/VCO2AT.

Authors’ response: Indeed, many different terminologies are used within the literature to define CPET-derived thresholds. We prefer to use the term ventilatory anaerobic threshold (VAT) over anaerobic threshold (AT), as is it makes more clear that the thresholds are derived from respiratory gasses (as opposed to i.e., lactate measurement). Nevertheless, if the reviewer or the editor insist on using AT instead, we are willing to adjust the terminology throughout the manuscript. 

Discussion :

Re-states main findings and discusses implications in context . Reasonable and moderate conclusions

Reviewer #2: 

This study make nonsense. If the aim is to establish whether the averaging interval of VO2 data has an impact in the VO2max value and the medical decisions adopting depending on this value why are the authors analyzing all results using 30-s averages? 

Authors’ response: We thank the reviewer for the critical appraisal of our study but we regret that the reviewer qualifies our study as nonsense. Before we can reflect on the reviewers points, we must emphasize that we did not analyze all results using 30-second data-averaging. In addition, we did not aim to establish whether data-averaging intervals impact VO2peak values alone. Instead, we assessed many more variables (as outlined on page 10) using 5 different data averaging intervals. Therefore, the point the reviewer merely refers to the determination of VO2peak values. 

Regarding the determination of VO2peak values, let us explain why we chose to analyze VO2peak values over an interval close to 30-seconds regardless of the used data-averaging method. In most (if not all) software packages used for CPET interpretation, VO2peak is calculated over a period that has to be manually set. That is, instead of using one point that is allocated as VO2peak, a lower limit and upper limit have to be set in order to demark the period over which VO2peak is calculated (see also Figure 1, red shaded area in each graph). With this in mind, the method chosen in the current study mimics how VO2peak is estimated in clinical practice. Nevertheless, we do agree with the reviewer that by doing so, the variability introduced by the data-averaging interval is attenuated as was already discussed on page 21, line 344-347. Nevertheless, this also reflects routine practice. 

Moreover, the metabolic cart data are incomplete, please add PETO2 and PETCO2 data.

Authors’ response: We are not sure what the reviewer is referring to as we did not disclose any metabolic cart data. We did however include the raw data on which the analyses are based. Is that what the reviewer is referring to? 

Are all metabolic carts used disclosed? Or all test were performed with the same metabolic cart?

Authors’ response: All tests are performed on the same metabolic cart as reported in the methods section in the paragraph “Preoperative cardiopulmonary exercise testing” on page 7 and 8. We included a data file with the raw data that was used for the analysis. We did not include the metabolic cart data as we believe that due to the relative small study sample it would be hard to guarantee anonymity of our participants. Nevertheless, on reasonable request the metabolic cart data could be shared after personal communication with the authors. 

Please add as as an example the full output data of one of the incremental exercise test with breath by breath data and then the calculated 10, 20 or 30 s averaged values.

Authors’ response: We added an example of a graphical display of the metabolic cart of patient 21 as Supporting information S3.

---

## [Decision Letter · Decision Letter 2]

3 Mar 2023

Influence of different data-averaging methods on mean values of selected variables derived from preoperative cardiopulmonary exercise testing in patients scheduled for colorectal surgery.

PONE-D-22-28377R2

Dear Dr. franssen,

We’re pleased to inform you that your manuscript has been judged scientifically suitable for publication and will be formally accepted for publication once it meets all outstanding technical requirements.

Kind regards,

Lindsay Bottoms

Academic Editor

PLOS ONE

Additional Editor Comments (optional):

Reviewers' comments:

Reviewer's Responses to Questions

**Comments to the Author**

1. If the authors have adequately addressed your comments raised in a previous round of review and you feel that this manuscript is now acceptable for publication, you may indicate that here to bypass the “Comments to the Author” section, enter your conflict of interest statement in the “Confidential to Editor” section, and submit your "Accept" recommendation.

Reviewer #3: All comments have been addressed

Reviewer #4: All comments have been addressed

2. Is the manuscript technically sound, and do the data support the conclusions?

Reviewer #3: Yes

Reviewer #4: Yes

3. Has the statistical analysis been performed appropriately and rigorously? 

Reviewer #3: Yes

Reviewer #4: Yes

4. Have the authors made all data underlying the findings in their manuscript fully available?

Reviewer #3: Yes

Reviewer #4: Yes

5. Is the manuscript presented in an intelligible fashion and written in standard English?

Reviewer #3: Yes

Reviewer #4: Yes

6. Review Comments to the Author

Reviewer #3: Much more focused paper, which addresses a scientific question well. My concerns have been addressed by the authors.

Reviewer #4: I have no further comments.

The authors have addressed my previous points satisfactorily - thank you for that.

7. PLOS authors have the option to publish the peer review history of their article (what does this mean?). If published, this will include your full peer review and any attached files.

Reviewer #3: No

Reviewer #4: **Yes: **Daniel Muniz Pumares

---

## [Editor Report · Acceptance letter]

8 Mar 2023

PONE-D-22-28377R2 

Influence of different data-averaging methods on mean values of selected variables derived from preoperative cardiopulmonary exercise testing in patients scheduled for colorectal surgery 

Dear Dr. Franssen:

I'm pleased to inform you that your manuscript has been deemed suitable for publication in PLOS ONE. Congratulations! Your manuscript is now with our production department. 

Kind regards, 

on behalf of

Dr. Lindsay Bottoms 

Academic Editor

PLOS ONE